# Learning Towards Emergence: Paving the Way to Induce Emergence by Inhibiting Monosemantic Neurons on Pre-trained Models

## Abstract

Emergence, the phenomenon of a rapid performance increase once the model scale reaches a threshold, has achieved widespread attention recently. The literature has observed that monosemantic neurons in neural networks gradually diminish as the model scale increases. Subsequently, *Learning From Emergence* is proposed to actively inhibit monosemantic neurons in relatively small neural networks (e.g., BERT and Swin-Transformer) for promoting model performance with fine-tuning. However, to ultimately achieve emergence, it is demanding to support the monosemantic neuron inhibition in the pretraining phase of large-scale models. Thus, this work further pushes the boundary of this research direction to be *Learning Towards Emergence (L2E)* and enables the training and validating of the impact of inhibiting monosemantic neurons on larger pre-trained neural networks (e.g., Pythia-70M, 410M, and 2.8B). More specifically, to bridge the gap in current research, we first conduct experiments on models of various scales (up to 6.9B) to validate the monosemantic ideas. Then, we present a novel method L2E to address the inefficient monosemantic neuron retrieval and ineffective monosemantic neuron inhibition when existing methods are applied in the pretraining phase of large-scale models. It employs an adjustable thresholding technique for efficient neuron retrieval, incorporates a False Killing Rate metric to assess inhibition effects, and proposes a regularization-style inhibition approach, which addresses the limitations of previous approaches in both efficiency and effectiveness. Experimental results demonstrate the effectiveness of L2E's monosemantic neuron inhibition and its efficiency in implementation with large-scale models.

## 1 Introduction

The success of large-scale pretraining models, such as GPT-3.5 (Ouyang et al., 2022), has drawn widespread attention in understanding their dynamics across different scales. Studies on Scaling Laws (Henighan et al., 2020; Kaplan et al., 2020) have analyzed the relationship between scale and performance, which typically follows a mild power law. However, recent research has observed dramatic performance improvements that defy these scaling laws when the model scales reach certain thresholds—a phenomenon termed *Emergence* (Wei et al., 2022). The resulting emergent abilities of these models are somehow recognized as a key factor in their success, prompting numerous follow-up investigations (Hu et al., 2024). Some studies suggest that the impressive emergence phenomenon may simply caused by deficiencies in observing and evaluating the accumulation of abilities (Schaeffer et al., 2023; Lu et al., 2024). But such deficiencies may persist for a long time because the commonly used unsupervised losses and weakly labeled datasets. Subsequently, a series of studies try to predict (Hu et al., 2024) and induce (Wang et al., 2024; Yan et al., 2024) emergence.

In earlier years, researchers propose the concept of monosemantic neurons to interpret model functionality (Bau et al., 2020; Elhage et al., 2022). These neurons form 1-to-1 mappings with human-friendly features (such as "dog" in images (Olah et al., 2020) or "Python" in code languages as shown in Figure 1(a)). In contrast, polysemantic neurons are activated for multiple features (Goh et al., 2021; Bricken et al., 2023) (see Figure 1(b)). The discovery of monosemantic neurons, especially when visualizing impressively (Olah et al., 2020), greatly excite researchers when neural networks are considered as black-box. However, following the favor of monosemantic neurons in

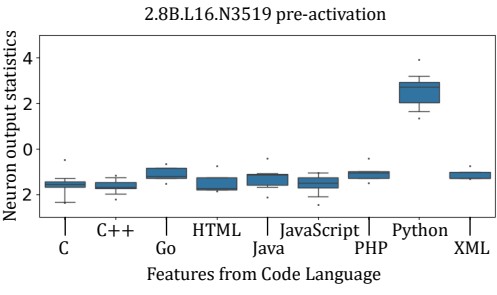 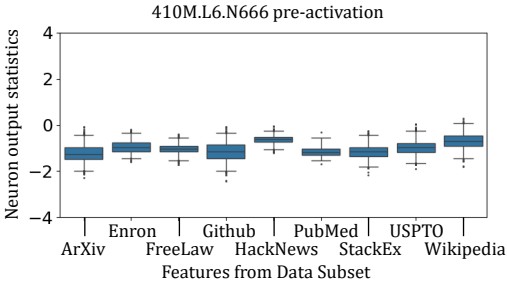

(a) The output statistics of monosemantic "Python" neuron on Code Language dataset. The neuron is in the layer 16, number 3519 of the Pythia 2.8B model.

(b) The output statistics of randomly selected neuron on Data Subset dataset. The neuron is in the layer 6, number 666 of the Pythia 410M model.

Figure 1: A demonstration of the concept "monosemantic". The left figure shows the output statistics of a monosemantic neuron, which is activated only by the feature "Python". This contrasts with a randomly selected neuron in the right figure. We use sparse probing (Gurnee et al., 2023) on Pythia models (Biderman et al., 2023) to detect monosemantic neurons.

explanation, it has become harder to detect monosemanticity as model scale increases (Huben et al., 2024; Gurnee et al., 2023; Bricken et al., 2023). Existing works also find that monosemantic neurons have less impact on model performance of larger models (Gurnee et al., 2023).

Based on these observations, Wang et al. (2024) hypothesize that the decrease of monosemantic neurons as a key factor towards better performance behind the increasing model scale, then propose *Learning From Emergence* to improve performance by actively inhibiting monosemantic neurons during the fine-tuning stage of relatively small models ($\leq$88M). To achieve that, Monosemanticity Score (MS) has been devised to quantify monosemanticity throughout the model training, which contrasts with literature that depend on specially labeled detection datasets and can only detect monosemantic neurons after training on frozen models (Gurnee et al., 2023; Huben et al., 2024). But Learning From Emergence is still in its early stages of exploration and has unresolved limitations (Yan et al., 2024). The validity of the MS metric and the above monosemanticity hypothesis lacks thorough investigation, as existing studies have not provided abundant experimental support. Moreover, the inhibition method faces challenges in effectiveness and efficiency when inhibiting monosemantic neurons in large-scale neural networks during the pretraining phase.

To explore the impact of inhibiting monosemantic neurons on the model performance, we further push the boundary of Learning From Emergence to *Learning Towards Emergence (L2E)* to induce emergence by inhibiting monosemantic neurons during pretraining on larger ($\times$30) models.

In this work, we first conduct an analysis to facilitate the understanding of monosemanticity. As monosemanticity is difficult to define explicitly (Olah et al., 2020; Elhage et al., 2022), we cross-validate the effectiveness of MS using carefully selected monosemantic neurons (Gurnee et al., 2023). Additionally, we perform an in-depth analysis of monosemanticity across different scales of models (from 70M to 6.9B). After validating the monosemantic idea, we propose L2E to enable monosemanticity inhibition for large-scale pretraining. More specifically, we first apply an adjustable thresholding technique to enable efficient monosemantic neuron retrieval. Then, we introduce the False Killing Rate as a metric to quantify the side effects of different inhibition levels and capture consistent patterns for guidance across scales. Finally, we propose a regularization-style inhibition approach, which addresses the ineffectiveness of existing work when applied to pretraining tasks. Experiments conducted on various tasks and scales using Pythia models (Biderman et al., 2023) (from 70M to 2.8B), validating the effectiveness and efficiency of L2E.

## 2 BACKGROUND

### 2.1 NEURON, AND ACTIVATED NEURON

A neural network can be viewed as multiple layers connected in series and parallel. Subsequent layers are computed as functions of previous layers, contributing to a differentiable and updatable

output. To understand the dynamics of networks, existing works zoom into individual layers and further study their "neurons" (Olah et al., 2020; Gurnee et al., 2023). Specifically, each layer consists of a set of neurons $\Theta = \{\theta\}$, where each neuron $\theta$ is a function that maps input $\mathbf{x}$ to an output value $z = \theta(\mathbf{x})$, where $\mathbf{x} \in \mathbb{R}^d$.

Within a neural network, the nonlinearity of neurons is primarily based on activation functions. ReLU$(z) = \max(z, 0)$ has become one of the most popular activation functions due to its simplicity and effectiveness (Glorot et al., 2011). While it outputs a constant 0 for negative inputs, its (largely) positive output is commonly recognized as "activated" (Mirzadeh et al., 2024). As research on activated neurons progressed, the concept is generalized, referring to any neuron output that significantly differs from its typical value, e.g., the neuron in Figure 2 for English. Besides, the position of studied neurons are no longer restricted, such as pre-activations (Gurnee et al., 2023) and class logits (Olah et al., 2017). Such an extension is useful for studying the dynamics of the whole networks (Wang et al., 2024).

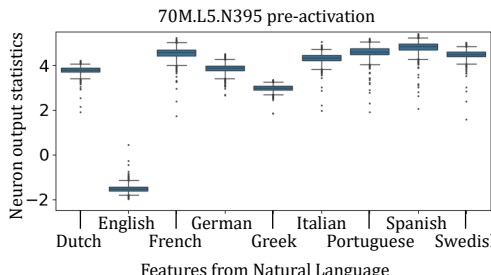

Figure 2: A monosemantic neuron with a negative mean difference. The average value of the neuron is also much larger than 0.

However, despite its widespread use as an intuitive concept, "activated neuron" remains challenging to define explicitly (Gurnee et al., 2023; Belinkov, 2022; Wang et al., 2024). Given a set of inputs $X = \{\mathbf{x}^{[i]}\}$, a neuron $\theta$ is considered activated for an input $\mathbf{x}^{[i]}$ if $z^{[i]} = \theta(\mathbf{x}^{[i]})$ has a significant deviation from its mean $\bar{z} = 1/|X| \sum_i z^{[i]} = 1/|X| \sum_i \theta(\mathbf{x}^{[i]})$, where it is hard to reach a consensus on the "significant" or the "deviation". Fortunately, to help understand and interpret the network, it is a reasonable and impressive approach to collect and demonstrate the statistics of neuron outputs.

## 2.2 STUDIES ON MONOSEMANTICITY

To conduct analysis for the monosemanticity of neurons, researchers propose human-friendly feature datasets. Formally, labeled feature datasets collect sets of input instances $X = \{\mathbf{x}^{[i]}\}$, each input is mapped to one of several labeled features $L = \{\ell^{[i]}\}$ (Gurnee et al., 2023; Bricken et al., 2023). For example, the natural language of Europarl documents contains $> 28k$ instances belonging to 9 labeled features (Gurnee et al., 2023). Given a neuron $\theta$, by feeding input $\mathbf{x}$ into the model, one can obtain the statistics of neuron values labeled by features. Specifically, we denote the output values $z$s with input of feature $\ell$ as:

$$C(\theta, \ell) = \{\theta(\mathbf{x}) : \mathbf{x}'s \text{ label is } \ell\}.$$

Based on the statistics or further transformations of the values, one can analyze the monosemanticity of each neuron. For example, a monosemantic neuron is expected to have a large mean difference for a feature. Additionally, when using sparse autoencoders as probing classifiers, it should achieve a high autointerpretability score or a high F1 score in predicting a feature (Gurnee et al., 2023; Bricken et al., 2023; Huben et al., 2024). Sparse probing is used as a tool for analysis in this paper.

However, probing experiments are time-consuming, making it crucial to develop alternative methods to boost the study of monosemanticity. Gurnee et al. (2023) first proposed estimating the neuron monosemanticity based on input weight norm and bias term, which is non-universal as not all models have bias terms (Yan et al., 2024). Further, Monosemanticity Score (MS) is proposed to dynamically analyze the monosemanticity based on sparsity (Wang et al., 2024). To be more specific, given a set of inputs $\{\mathbf{x}^{[i]}\}_{i=1}^n$ and the corresponding outputs of a neuron $\{z^{[i]}\}_{i=1}^n$, MS is defined as:

$$\phi(z^{[i]}) = \frac{(z^{[i]} - \bar{z})^2}{S^2}, \tag{1}$$

where $\bar{z}$ is the mean of $\{z^{[i]}\}$, and $S^2$ is the sample variance. During model training, only samples before $z^{[i]}$ are observable. In this case, the MS can be calculated incrementally in linear time complexity (Wang et al., 2024). Given these advantages, Wang et al. (2024) proactively inhibited monosemanticity based on the MS. However, the effectiveness of the MS metric lacks experimental validation (Yan et al., 2024).

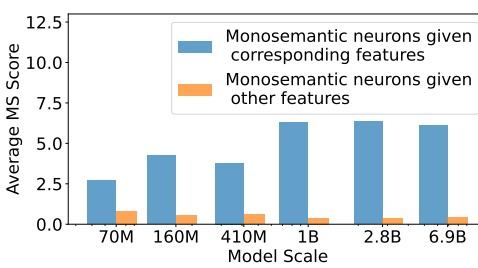 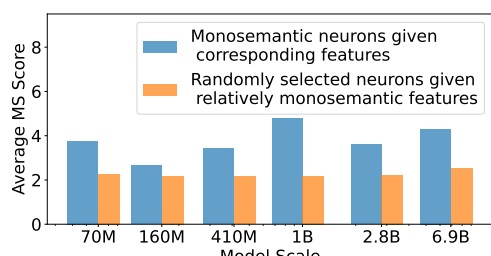

(a) Given the most monosemantic neurons, the average MS scores when the input contains monosemantic features $\phi_\ell$ (blue) or does not contain monosemantic features $\phi_\ell^-$ (orange).

(b) The average MS of monosemantic neurons (blue) given corresponding features $\phi_\ell$ compared with randomly selected neurons (orange) given relatively monosemantic features $\phi_{\ell*}$.

Figure 3: Validation of the effectiveness of MS. We probe neurons in Pythia models (Biderman et al., 2023) based on feature datasets Code Language (a) and Data Subset (b) (Gurnee et al., 2023).

## 3 Metric Validation and Analysis

The MS evaluation metric is proposed to analyze monosemanticity efficiently and dynamically (Wang et al., 2024; Yan et al., 2024). However, its effectiveness has not been throughly verified experimentally. The hypothesis that monosemanticity is negatively correlated with increasing scale has been only tentatively validated by deactivating monosemantics neurons and examining increased losses (Gurnee et al., 2023; Wang et al., 2024). Therefore, we *i)* cross-validate MS using monosemantic neurons detected by probing method (Belinkov, 2022), and *ii)* conduct analysis on monosemanticity based on MS in this section.

### 3.1 Validation of Monosemanticity Score

As introduced in Section 2, it is hard to explicitly define monosemanticity (Wang et al., 2024; Yan et al., 2024). To validate that the MS metric can indeed reflect the monosemanticity of neurons, we choose the sparse probing (Gurnee et al., 2023) as a cross-validation for evaluating MS from two perspectives: *i)* For monosemantic neurons, their MS values differ significantly when given inputs with specific features compared to other inputs; *ii)* When considering the most monosemantic feature, MS can effectively distinguish monosemantic neurons from others.

**MS Can Detect Activated Monosemantic Neurons.** Firstly, we compare the MS values of monosemantic neurons when given and not given the corresponding features. The top 10 monosemantic neurons are selected by sparse probing (Gurnee et al., 2023) on Pythia models across scales (70M to 6.9B) (Biderman et al., 2023). To be more specific, given a set of inputs $\{\mathbf{x}^{[i]}\}_{i=1}^n$ and a monosemantic neuron $\theta$ with corresponding feature $\ell$, its output values $Z = \{z^{[i]}\}_{i=1}^n = \{\theta(\mathbf{x}^{[i]})\}_{i=1}^n$ can be partitioned as $C_\ell = C(\theta, \ell)$ and $C_\ell^- = \cup_{\ell' \neq \ell} C(\theta, \ell')$. We can calculate the MS values of $Z$ within $C_\ell$ and $C_\ell^-$ respectively, denoting the mean of each set as $\phi_\ell$ and $\phi_\ell^-$:

$$\phi_\ell = \frac{\sum_{z \in C_\ell}(z - \bar{z})^2}{|C_\ell|S^2}, \phi_\ell^- = \frac{\sum_{z \in C_\ell^-}(z - \bar{z})^2}{|C_\ell^-|S^2}. \tag{2}$$

Intuitively, a monosemantic neuron should be activated when given the inputs from corresponding feature, thus a larger $\phi_\ell$ if MS is effective. Based on the top-10 monosemantic neurons, we derive $\phi_\ell$ and $\phi_\ell^-$ based on the Code Language feature dataset (Figure 3(a)), where results for two more datasets are provided in Figure 7 in the Appendix. It is clear that the MS values of neurons with monosemantic featuress are significantly different from those without monosemantic features. This indicates that MS is sensitive to monosemanticity when the corrsponding features are given.

**MS is Non-sensitive to Non-monosemantic Neurons.** As MS is effective for monosemantic neurons, it is also important that MS should be insensitive to non-monosemantic neurons. To validate this, across different scales, we randomly select 10 neurons in addition to the monoseman-

tic neurons. To compared with the score $\phi_\ell$ calculated for monosemantic neurons, we calculate the MS for those randomly selected neurons and mine their statistically more monosemantic features instead. Mathematically, for each feature $\ell^{[i]} \in L$, we calculate its average MS score $\phi_{\ell^{[i]}} = \sum_{z \in C_{\ell^{[i]}}} (z-\bar{z})^2 / |C_{\ell^{[i]}}| S^2$, where monosemanticity is higher when the score is higher. Thus, we denotes the feature $\ell^*$ with the highest $\phi_{\ell^{[i]}}$ as its relatively monosemantic feature, that is:

$$\ell^* = \max_{\ell \in L} \frac{\sum_{z \in C_\ell} (z - \bar{z})^2}{|C_\ell| \, S^2}. \tag{3}$$

The corresponding $\phi_{\ell^*}$ is used to compare with $\phi_\ell$ for monosemantic neurons. The results are shown in Figure 3(b), where $\phi_\ell$ and $\phi_{\ell^*}$ are derived from the Data Subset feature dataset. Additional results are given in Figure 8 in the Appendix. It is clear that the MS values of neurons with monosemantic features are significantly different from those from random neurons. This indicates that MS can effectively distinguish monosemantic neurons from other neurons.

## 3.2 ANALYSIS OF MONOSEMANTICITY BASED ON MS

Recall the assumption that monosemanticity is negatively correlated with increasing scale. It is proposed by existing work and preliminarily validated by turning off monosemantic neurons and observing the increased loss (Gurnee et al., 2023; Wang et al., 2024). In this subsection, we conduct further analysis using a fine grid of both scales (6 scales from 70M to 6.9B) and layers.

First, we use the Kolmogorov-Smirnov (K-S) Test to compare how outstanding the influence of the most monosemantic features $\ell^*$ are across models of different sizes. We randomly select 1000 neurons per scale of model. For each neuron, we provide inputs of different features and records their MS values. Based on this, we calculate the average score for each feature. Similar to equation 3, as monosemantic feature is unavailable for a randomly selected neuron, we treat the feature with the highest average score as the relatively monosemantic feature of the neuron $\ell^*$. For each scale of model, we can treat the MS scores from inputs in feature $\ell^*$ as a set of monosemantic samples $\phi_{\ell^*}$, which contrasts with the universal set, i.e., the

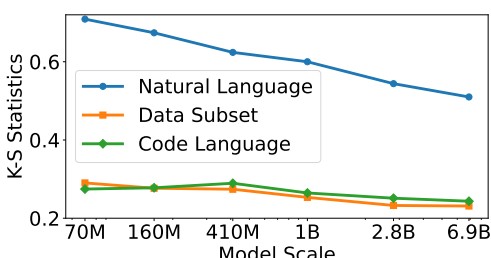

Figure 4: K-S test for the monosemanticity levels across model scales on 3 feature datasets.

MS scores of all the samples $\{\phi(z^{[i]})\}_{i=1}^{n}$. When a neuron is more monosemantic, the difference between the two sets of samples should be greater.

To obtain statistical significance, we apply the K-S test (Peacock, 1983) on the set of MS scores from relatively monosemantic features and the universal set. The K-S test, a widely used nonparametric hypothesis test, determines whether two sample sets originate from different distributions. In our experiment, we compare the K-S statistics, which is positive related the difference between the 2 sets. The results are shown in Figure 4, shown the results on 3 feature datasets Natural Language, Data Subset, and Code Language (Gurnee et al., 2023). The K-S statistics decrease as the scale increases, indicating that the prominence of the monosemantic set diminishes. These results validate that monosemanticity is negatively correlated with larger-scale models.

Additionally, we investigate monosemanticity across layers. With 1,000 randomly selected neurons for each scale, we calculate the MS values for each neuron and record the mean scores of its most monosemantic feature $\phi_{\ell^*}$ according to equation 3. As shown Figure 5 and 9, a clear drop in MS, indicating lower monosemanticity, can be observed as the scale increases.

## 4 LEARNING TOWARDS EMERGENCE

Recall that to inhibit monosemanticity for large scale models, current method lacks insightful design for effectiveness and is inefficient. In this section, we first dig into the dynamic of monosemanticity and propose an upgrade version named L2E to support effective pretraining with reasonable configurations. Then, we introduce an adjustable module to solve the efficiency bottleneck.

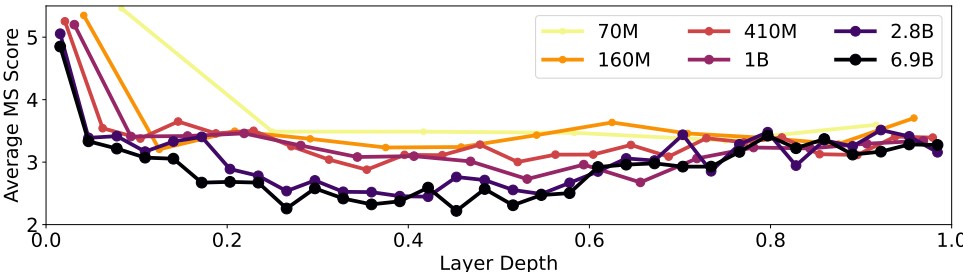

Figure 5: Statistics of MS across scales and layers. The results are obtained on Natural Language feature dataset (Gurnee et al., 2023). Larger models are of deeper colors, which can be clearly observed that their scores are smaller, indicating lower monosemanticity.

### 4.1 RECALL OF MEmeL

As discussed in Section 2, Wang et al. (2024) introduces the MEmeL method to first detect monosemantic neurons and retrieve them, then inhibit them to promote the relatively small neural networks. However, the current method is limited by following 3 limitations. First, monosemantic neuron detection is an $O(1)$ operation based on MS. Values of neurons $\mathbf{z}$ in a layer are ranked by their monosemantic scores $(\phi(\mathbf{z}))$. Then, the top-$k$ neurons are selected for inhibition, where $k$ is a relatively small number ($\leq 100$). But this setting of $k$ lacks solid justification. A deeper insight into the dynamics of monosemanticity is needed for high-quality detection. Secondly, top-$k$ neurons retrieval becomes inefficient when the number of neurons in a layer is large and $k$ increases. Specifically, the time complexity $O(kN)$ of comparing $N$ neurons in the layer increases to sorting with $O(N \log N)$. This creates a significant bottleneck as the scale increases, necessitating a more efficient method. Thirdly, after obtaining the neurons to deactivate, MEmeL proposed Reverse Deactivation to inhibit the selected neurons with theoretical guarantees. However, this method assumes that neuron activation is a well-trained result rather than a mistake due to insufficient training. This assumption is not always valid, especially when applied at the beginning of pretraining. Therefore, we propose L2E to address the 3 limitations of existing works on inhibiting monosemantic neurons.

### 4.2 L2E: LEARNING TOWARDS EMERGENCE

**False Killing Rate: Determines How Many Neurons to Inhibit.** Recall that the previous method majorly tried to validate the assumption of the influence of monosemanticity, which only inhibited no more than 100 most monosemantic neurons (Wang et al., 2024). However, large models has a great number of neurons in a single layer, e.g., 5,242,880 for the 2.8B pythia model, where the influence of inhibiting dozens of neurons is neglegable.

Here we discuss the intuition behind the setting of $k$. From the perspective of memorization and reasoning, monosemantic and polysemantic neurons are assumed to play different roles. To prevent the model from overfitting-style rote memory, we aim to keep the polysemantic neurons and inhibit the monosemantic ones. This is achieved by filtering out the neurons with top-$k$ MS values. A small $k$ will lead to weak inhibition, while a large $k$ may also inhibit polysemantic neurons, which impairs functionality. To quantify the impairment, we propose the False Killing Rate (FKR) to measure the proportion of unexpected inhibitions where the inputs are not from monosemantic features. To be more specific, given a dataset with $n$ input instances $X = \{\mathbf{x}^{[i]}\}_{i=1}^{n}$ and a layer of $N$ neurons $\mathbf{z} = \{z_j\}_{j=1}^{N}$, the FKR is defined as:

$$\text{FKR} = \frac{\sum_{i=1}^{n} \sum_{j=1}^{N} \mathbb{1}\left(\mathbf{x}^{[i]} \notin \ell_j^*\right) \cdot \mathbb{1}\left(\phi(z_j^{[i]}) \geq \tau_k\right)}{\sum_{i=1}^{n} \sum_{j=1}^{N} \mathbb{1}\left(\phi(z_j^{[i]}) \geq \tau_k\right)}, \tag{4}$$

where $z_j$ is the $j$-th neuron in the layer and $\ell_j^*$ is its relatively monosemantic feature as defined in equation 3. $\mathbb{1}(\cdot)$ is the indicator function and $\tau_k$ is the $k$-th largest MS value. The FKR measures the proportion of unexpected inhibitions ($\mathbf{x}^{[i]} \notin \ell_j^*$) when the inputs are not from monosemantic features. Ideally, we aim to reduce monosemantic neurons while preserving polysemantic ones. Therefore, we must balance between achieving sufficient inhibition and maintaining a low FKR.

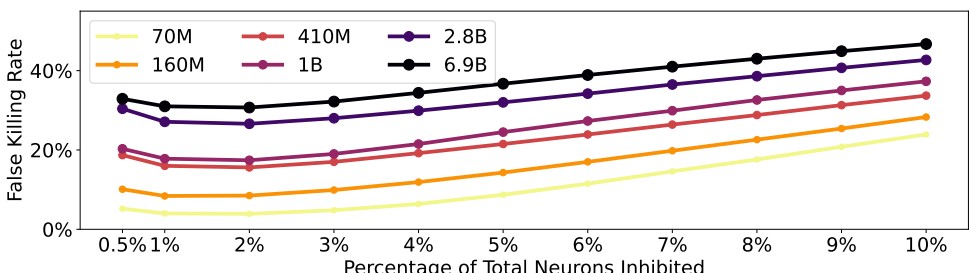

Figure 6: The False Killing Rate across 6 scales of Pythia models (Biderman et al., 2023) on Natural Language, where analysis on other feature datasets are given in Appendix B.1. The x-axis represents the percentage of neurons inhibited. The red line marks the empirical optimal $k$ with the lowest FKR.

To meet our goal, we conduct experiments on Pythia models of different scales (Biderman et al., 2023). Interestingly, despite the overall positive relationship, the FKR initially decreases as the number of inhibited neurons increases. We find a consistent trend when the number of inhibited neurons is determined based on percentage, as shown in Figure 6. One can easily observe an ideal $k$ (i.e., 2% of the total number of neurons) that minimizes the FKR. This empirical setting remains consistent across different scales, which is a promising result for large-scale models. We further validated this setting with experiments in Appendix B.3.

Moreover, the FKR is increasing when the scales of the models increase. This further validate the proposation that larger models are more polysemantic, so that inhibiting neurons will more likely lead to false killing. A more insight analysis is given in Appendix B.2.

**Efficient Neuron Retrieval.** When the inhibition level comes to several percentages (e.g., 2%) of neurons in a layer, efficiency becomes a bottleneck. Originally, finding the largest MS value costs $\Theta(N)$ times for a layer of $N$ neurons, which will be extended to $\Theta(kN) \sim \Theta(N^2)$ for the top-$k$ inhibition. Using special data structures such as heap can reduce the cost to $O(N \log(k)) \sim O(N \log(N))$, which is still expensive for large scale models.

Here, we design a moving threshold to circumvent the calculation of precise top-$k$ value. Being inspired while conducting monosemanticity analysis, we found that the inhibition should be better as a global percentage rather than an in-batch ranking. For example, a single batch of inputs may activate fewer monosemantic neurons, which should be given a lower level of inhibition. Therefore, we maintain a moving threshold to inhibit the most monosemantic neurons globally. The detailed implementation, primarily an engineering design, is provided in Appendix A. Our design involves only an element-wise comparison per batch with an $O(1)$ update to converge the threshold to the global value of the 2%-th highest MS, achieving efficient inhibition.

**Efficient Neuron Inhibition.** Given the identified monosemantic neurons, the previous method developed Reverse Deactivation to inhibit them (Wang et al., 2024). This approach makes use of the model's reliance on monosemanticity to naturally deactivate neurons. In short, it assumes that a monosemantic neuron is well-trained and effective, such that reducing its activation would lead to an increase in loss. However, the assumption is more valid for a well-trained model but less prevalent in the pretraining stage, particularly at its very beginning. To handle the problem, we propose a regularization-style method to encourage the inhibition. Specifically, we introduce a new term to the loss function to penalize the MS values of the monosemantic neurons directly, that is, to minimize equation 1 for each selected neuron $\theta$ and input $\mathbf{x}^{[i]}$:

$$\min_{\boldsymbol{\omega}} \frac{(\theta_{\boldsymbol{\omega}}(\mathbf{x}^{[i]}) - \bar{z})^2}{S^2},$$

where $\boldsymbol{\omega}$ are the trainable parameters of neuron $\theta$. However, during implementation, we discovered that the denominator term $S^2$ could become extremely small, leading to unstable gradients. We turn to a logarithmic transformation to stabilize the gradients, minimizing the following term instead:

$$\min_{\boldsymbol{\omega}} \log \frac{(\theta_{\boldsymbol{\omega}}(\mathbf{x}^{[i]}) - \bar{z})^2}{S^2} = \min_{\boldsymbol{\omega}} \log (\theta_{\boldsymbol{\omega}}(\mathbf{x}^{[i]}) - \bar{z})^2 - 2 \log S,$$

Table 1: The main results of applying L2E to inhibit 2% neurons of 2 middle layers of each Pythia models, where best results are in bold font.

| Setting | | 0-shot | | | | 5-shot | | | |
| --- | --- | --- | --- | --- | --- | --- | --- | --- | --- |
| Datasets | | ARC-C | PIQA | SciQ | ↑ | ARC-C | PIQA | SciQ | ↑ |
| **70M** | Pythia | 0.1706 | 0.5887 | 0.6430 | - | 0.1834 | 0.5843 | 0.4050 | - |
| | Dropout | 0.1681 | 0.5930 | 0.6350 | -0.4% | 0.1741 | 0.5925 | 0.4110 | 0.4% |
| | L2E | **0.1877** | **0.5963** | **0.6510** | **2.3%** | **0.1860** | **0.6034** | **0.4380** | **4.7%** |
| **410M** | Pythia | 0.1852 | 0.6376 | 0.7400 | - | 0.1988 | 0.6415 | 0.4850 | - |
| | Dropout | **0.2090** | 0.6289 | **0.7520** | **1.7%** | **0.2056** | 0.6344 | 0.4820 | -0.2% |
| | L2E | 0.2031 | **0.6398** | 0.7470 | **1.7%** | 0.2039 | **0.6518** | **0.4870** | **1.3%** |
| **2.8B** | Pythia | 0.2253 | 0.6768 | 0.7910 | - | 0.2346 | **0.6844** | 0.4810 | - |
| | Dropout | 0.2167 | 0.6763 | 0.8130 | 0.8% | 0.2355 | 0.6768 | **0.4910** | 0.2% |
| | L2E | **0.2304** | **0.6795** | **0.8150** | **1.9%** | **0.2415** | 0.6817 | **0.4910** | **1.0%** |

where the second term is a constant and can be ignored. So that:

$$\min_{\boldsymbol{\omega}} \mathcal{L}_{MS} = \min_{\boldsymbol{\omega}} \log \left( \theta_{\boldsymbol{\omega}}(\mathbf{x}^{[i]}) - \bar{z} \right)^2, \tag{5}$$

where the term $\mathcal{L}_{MS}$ can be added to the loss function. It offers a straightforward approach to reducing monosemanticity and is general to models at any stage of training.

In summary, L2E is designed for effectiveness and efficiency monosemanticity inhibition for pretraining in large-scale models. It builds on the dynamics of monosemanticity and introduces the False Killing Rate, which guides us in determining the optimal number of neurons to inhibit. A moving threshold is proposed for efficient identification of the most monosemantic neurons. Besides, it develops a regularization-style approach to encourage the inhibition of monosemantic neurons, addressing previous shortcomings in pretraining.

## 5 EXPERIMENT

In this section, we evaluate the effectiveness and efficiency of L2E on large-scale pretraining tasks. First, we introduce the experimental settings. Then, we compare L2E's performance with baseline methods. Finally, we discuss the limitations of our study.

### 5.1 EXPERIMENTAL SETTINGS

**Backbone Model.** Limited by our computational resources ($3 \times 8$ H100 GPUs), we choose Pythia (Biderman et al., 2023) as the backbone model in this paper. Pythia is proposed across various sizes, targeting research for scaling understanding—which perfectly fits our requirements. For configuration details, we adopt the same hyperparameters as the original Pythia models, including learning rate, batch size, and optimizer. We also use the deduplicated Pile training data (Gao et al., 2021a) that is indexed and available on the repository of Pythia for consistency. The only difference is that our total training steps are 10 percent of the original paper (14.3k versus 143k), trying to demonstrate the empirical results within accessible GPUs. We test three model scales: 70M, 410M, and 2.8B to capture MEmeL's impact across different sizes. Because of our limited resources, other sizes will be included in the future. We use the widely used evaluation tools for large models, LM Evaluation Harness (Gao et al., 2021b), to test multiple datasets on our model.

**L2E Settings.** The following main experiments, we inhibit neurons within the middle two layers, which are previously hypothesized and analyzed to be more polysemantic (see discussions in Section 3.1 and Appendix B.2). In each layer, we apply L2E at the output of each transformer block. More discussions of inhibiting other layers are presented in Appendix B.5. The MEmeL loss term is added to the original loss function with weights of 1e-9, 1e-10, and 1e-11 for Pythia 2.8b, 410m, and 70m, respectively, to prevent it from dominating the overall loss. Following the analysis in Section 4.2, we set the number of neurons to inhibit at 2% of the total neurons in each layer. More experiments on different inhibition rates are discussed in Appendix B.3.

## 5.2 MAIN EXPERIMENT RESULTS

In Table 1, we report the accuracy of applying L2E to inhibit monosemantic neurons in three scales of Pythia models. We evaluate the models on three datasets: ARC-Challenge (arc_c) (Clark et al., 2018), PIQA (Bisk et al., 2020), and SciQ (Welbl et al., 2017). Both 0-shot and 5-shot results are reported. We also add a Dropout baseline with 0.2 randomly drop. Our findings show that L2E consistently outperforms the original models across all datasets and scales. In the zero-shot setting, L2E achieves 2.0% higher accuracy than Pythia and 1.3% than Dropout on average. On the other hand, L2E shows better average improvements in the few-shot scenario (2.3% and 2.3% higher accuracy). These results clearly demonstrate L2E's effectiveness in enhancing large-scale pretraining models. More experimental results are provided in Appendix B.4B.6.

## 5.3 EFFICIENCY ANALYSIS

Here, we compare the efficiency of L2E with the MEmeL method and the original Pythia model. Table 2 shows the results, recorded as the average time cost per step. At the 70M scale, both MEmeL and L2E show mild cost increases, consistent with (Wang et al., 2024). However, as discussed in Section 4.2, with a complexity of $O(N \log N)$, the time cost escalates significantly as the scale in-

Table 2: The time cost (ms) of Pythia, MEmeL, and L2E. # Param records the number of parameters per layer.

| Scales | 70M | | 410M | | 2.8B | |
|---|---|---|---|---|---|---|
| # Param | 1,048,576 | | 2,097,152 | | 5,242,880 | |
| **Models** | Time | ↑ | Time | ↑ | Time | ↑ |
| Pythia | 605.8 | - | 2383.1 | - | 11285.1 | - |
| MEmeL | 664.8 | 9.7% | 2727.3 | 14.4% | 14330.8 | 27.0% |
| L2E | 626.5 | 3.4% | 2432.2 | 2.1% | 11451.7 | 1.48% |

creases. When a scale of 2.8B ($\times 5$ parameters per layer), the additional time cost ratio for MEmeL jumps from 9.7% to 27.0%. In contrast, our L2E's ratio even decreases from 3.4% to 1.5%. This reduction is due to L2E's element-wise operations being amortized by the superlinear cost of the Attention framework as the number of parameters increases, aligning with our analysis in Section 4.2.

## 5.4 LIMITATIONS

There is still work to be done to fully validate the effectiveness of L2E, especially its potential to induce Emergence. Current experiments are limited to Pythia models, with the largest size being 2.8B and training steps at 10% of full pretraining. This limitation is an unavoidable trade-off at this trial stage, where detailed analysis is required with our resources fully utilized.

Additionally, monosemantic analysis relies heavily on feature datasets, which significantly influence the results. Our analyses show particular consistency with the Natural Language feature dataset, a high-quality set where different languages are clearly distinguished. For each language, identifying its context neurons is straightforward, with an inference accuracy of 100% (Gurnee et al., 2023). However, there's no consensus on how to properly define feature datasets, which hinders comprehensive monosemanticity analysis. To validate the universality of our findings, one need to conduct analysis using more high-quality datasets. See Appendix B.1 for more results and discussions.

## 6 CONCLUSION

Our paper addresses the challenges in understanding and improving large-scale pretraining models through the lens of monosemanticity. We experimentally validate the metric Monosemanticity Scale for quantifying monosemantic levels, which further enables a comprehensive analysis of monosemanticity dynamics across different model scales. Our main contribution, L2E (Learning Towards Emergence), offers a novel approach to inhibiting monosemantic neurons in large-scale pretraining models. By incorporating the False Killing Rate metric, employing an adjustable thresholding technique, and proposing a regularization-style inhibition approach,L2E addresses the limitations of previous methods in both efficiency and effectiveness. Experimental results on Pythia models across various tasks and scales demonstrate the potential of L2E in enhancing model performance during pretraining. This work contributes to the ongoing research on understanding and inducing emergence in large language models, paving the way for future advancements in the field.

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

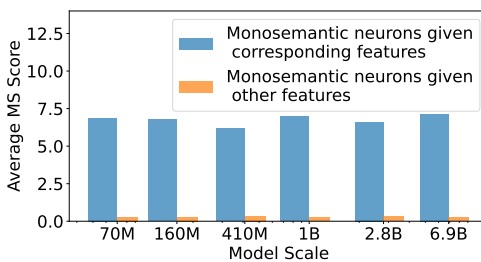
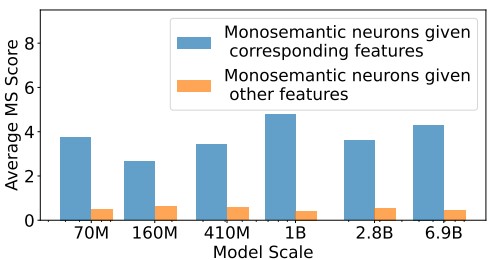

(a) Results on Natural Language feature dataset.  (b) Results on Data Subset feature dataset.

Figure 7: Additional validation results for the effectiveness of MS on monosemantic neurons.

# A  MOVING THRESHOLD

To enable efficient computation and capture a global impact, we maintain a moving threshold for the top-$k$ inhibition. To be more specific, in the first several batches, we warm up the threshold $\tau^*$ by precisely calculating the $k$-th largest MS values and recording the mean. After the warm-up, for each incoming batch, we inhibit the neurons with MS values larger than $\tau^*$, which is a simple element-wise comparison. To dynamically update $\tau^*$, we record the number of inhibited neurons $k^*$ and update it accordingly:

$$\tau^* \leftarrow \tau^* + \frac{k^* - k}{N}, \tag{6}$$

This will increase $\tau^*$ if the current inhibition level is too high, which has more inhibited neurons (large $k^*$), and vice versa. Such a negative feedback will push $\mathbb{E}[k^*]$ to $k$.

# B  ADDITIONAL EXPERIMENT RESULTS

## B.1  ADDITIONAL ANALYSIS RESULTS

Our experiments are conducted on the Pythia models (Biderman et al., 2023) with feature dataset from Gurnee et al. (2023). We use feature datasets Natural Language, Data Subset, and Code Language to validate the effectiveness of MS and the monosemanticity hypothesis. The statistics are given in Table 3, where Size is the number of inputs in each datasets,

Table 3: The statistics of feature datasets.

| Dataset | Size | Length | $|L|$ |
|---|---|---|---|
| Natural Language | 28084 | 512 | 9 |
| Data Subset | 8413 | 512 | 9 |
| Code Language | 5397 | 512 | 9 |

Length is the length of each input, and $|L|$ is the number of features. For more details, please refer to Gurnee et al. (2023).

In addition to the analysis shown in Section 3, more results are given here. The Figure 7 and Figure 8 are used to verify the effectiveness of MS when applied monosemantic neurons and other neurons. The Figure 9 demonstrates the MS changes across scales and layers.

Note that though most of the results are consistent with the main text, we still want to highlight some special outliers. For example, in Figure 8(b), the 70M model on the Code dataset has a lower MS from monosemantic neurons compared with those from others.

To find out the reason, we further analyze the probing results of these feature datasets. As in Gurnee et al. (2023), the most monosemantic neurons are probed based on F1 scores (using the neuron outputs to predict feature), we dis-

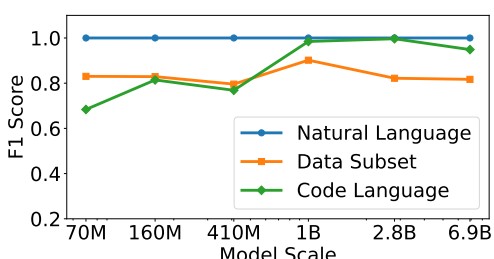

Figure 10: Average F1 scores of the top-10 neurons using sparse probing (Gurnee et al., 2023).

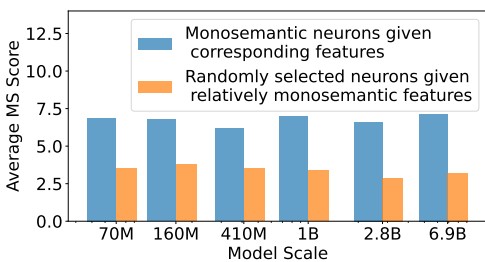 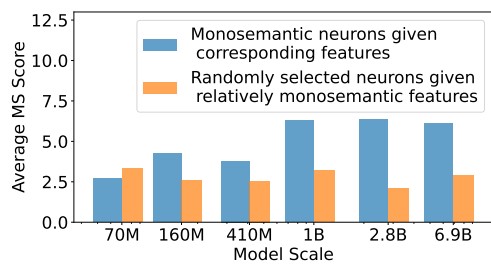

(a) Results on Natural Language feature dataset.  (b) Results on Code Language feature dataset.

Figure 8: Additional validation results for the effectiveness of MS in distinguishing monosemantic neurons from others.

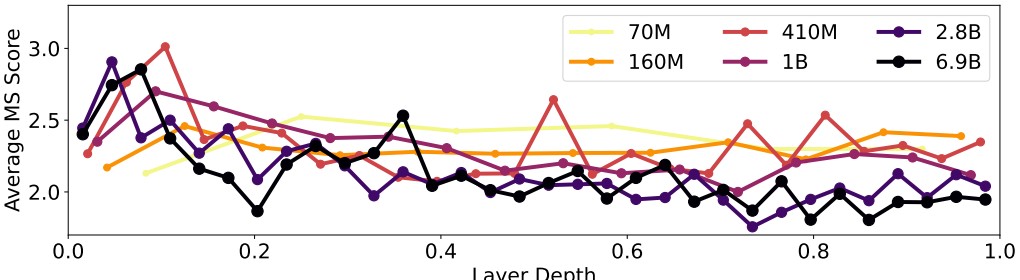

(a) Results on Data Subset feature dataset.

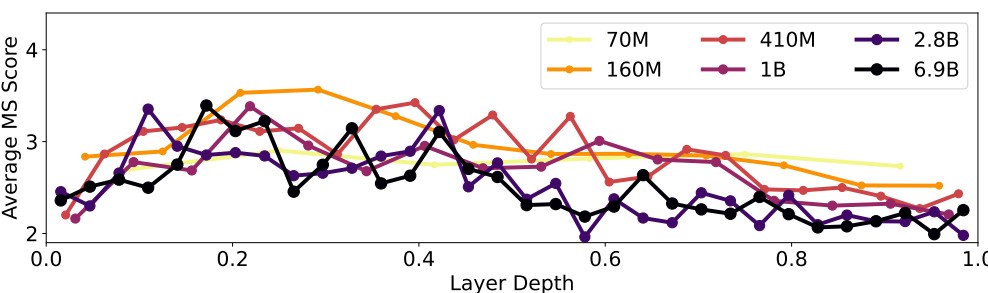

(b) Results on Code Language feature dataset.

Figure 9: Additional validation results support the trend of decreasing monosemanticity as model scale increases.

play the average F1 scores of the top-10 neurons in Figure 10. We find that the neurons of the 70M model have significantly lower scores on the Code Language dataset. This suggests that the probing classifier may not be effective in detecting monosemantic neurons in the 70M model. As a consequence, the results of 70M model on the Code Language dataset are also abnormal, such as Figure 8(b), Figure 9(b), and Figure 4.

These patterns further emphasize the importance of reliable feature datasets. According to Figure 10, only Natural Language consistently yields high F1 scores, owing to the inherent distinguishability of different languages. In contrast, when we inspect the results for Code Language with 9 features, 37 neurons have F1 scores $> 0.9$, with 21 (approximately 57%) being Python-related, while none are associated with HTML or XML. This disparity suggests either a biased distribution of the model's capabilities or a weakness in the feature dataset. In our analysis of inhibition levels (Figure 6 on Natural Language), a 1% inhibition on the 70M model leads to a False Killing Rate of 5.2%. However, when we examine the other two datasets, the FKR rises to 21.8% for Data Subset and even 67% for Code Language, failing to effectively distinguish monosemanticity for analysis. To further boost the study of monosemanticity, high-quality feature datasets are essential.

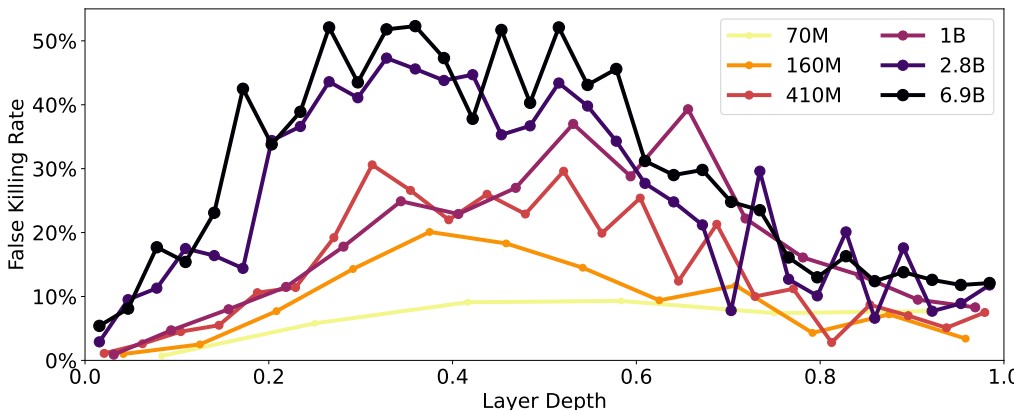

Figure 11: The False Killing Rate (FKR) varies across different layers in the Pythia models (Biderman et al., 2023) when the inhibition level is set to 2%. Larger models, represented by deeper colors, and middle layers exhibit higher FKR, suggesting they are more polysemantic.

Table 4: The results of applying L2E to 2 middle layers of each Pythia models with different levels of inhibition. The best results are in **bold** font where the second best are with underline.

| Setting | | **0-shot** | | | | **5-shot** | | | |
|---------|---------|--------|--------|--------|------|--------|--------|--------|------|
| Datasets | | ARC-C | PIQA | SciQ | ↑ | ARC-C | PIQA | SciQ | ↑ |
| **70M** | Pythia | 0.1706 | 0.5887 | 0.6430 | - | 0.1834 | 0.5843 | 0.4050 | - |
| | L2E-1% | **0.1903** | 0.5947 | **0.6750** | **4.1%** | **0.1869** | 0.5979 | 0.4350 | 4.0% |
| | L2E-2% | 0.1877 | **0.5963** | 0.6510 | 2.3% | 0.1860 | **0.6034** | **0.4380** | **4.7%** |
| | L2E-3% | 0.1817 | 0.5849 | 0.6430 | 0.5% | 0.1834 | 0.5892 | 0.4290 | 2.5% |
| **410M** | Pythia | 0.1852 | 0.6376 | 0.7400 | - | 0.1988 | 0.6415 | 0.4850 | - |
| | L2E-1% | 0.1928 | **0.6425** | 0.7300 | 0.2% | **0.2108** | 0.6464 | 0.4690 | 0.1% |
| | L2E-2% | 0.2031 | 0.6398 | **0.7470** | **1.7%** | 0.2039 | **0.6518** | **0.4870** | **1.3%** |
| | L2E-3% | **0.2108** | 0.6409 | 0.7290 | 1.2% | 0.2099 | 0.6442 | 0.4650 | -0.5% |
| **2.8B** | Pythia | 0.2253 | 0.6768 | 0.7910 | - | 0.2346 | **0.6844** | 0.4810 | - |
| | L2E-1% | 0.2295 | 0.6774 | 0.7930 | 0.4% | 0.2321 | 0.6839 | 0.4880 | 0.1% |
| | L2E-2% | **0.2304** | **0.6795** | **0.8150** | **1.9%** | 0.2415 | 0.6817 | **0.4910** | 1.0% |
| | L2E-3% | 0.2261 | 0.6763 | 0.8130 | 1.3% | **0.2517** | 0.6823 | **0.4910** | **1.8%** |

## B.2 ADDITIONAL VALIDATION OF MONOSEMANTICITY HYPOTHESIS

In addition to the analysis given in Subsection 3.2, we further validate the monosemanticity hypothesis by examining the False Killing Rate (FKR) across different layers in the Pythia models (Biderman et al., 2023). Using the aforementioned setting (2% of the total number of neurons in each layer), we analyze how FKR varies across layers, as shown in Figure 11. Larger models, represented by deeper blue colors, are more likely to experience false killing when inhibiting neurons. This aligns with our hypothesis that larger models are more polysemantic. Besides, middle layers exhibit higher FKR, suggesting they are more polysemantic. This coincides with their role in abstraction and reasoning. In contrast, the top and bottom layers, being closer to specific inputs and outputs, are inevitably more monosemantic.

## B.3 INHIBITING DIFFERENT AMOUNT OF NEURONS

In this section, we further investigate the impact of inhibiting different amounts of neurons on model performance. We compare our default inhibition level of 2% with 1% and 3% of the total number of neurons in each layer. The results are shown in Table 4. While all settings of our methods achieve

Table 5: The results of applying L2E to top and bottom layers (Top-Bot), 1/3 middle layers (Mid-1/3), and 2 middle layers (Mid-2) as settings in the main experiment. The best results are in **bold** font. (*) Note that Mid-2 and Mid-1/3 are the same for 70M model with 6 layers.

| Setting | | **0-shot** | | | | **5-shot** | | | |
|---|---|---|---|---|---|---|---|---|---|
| Datasets | | ARC-C | PIQA | SciQ | ↑ | ARC-C | PIQA | SciQ | ↑ |
| **70M** | Pythia | 0.1706 | 0.5887 | 0.6430 | - | 0.1834 | 0.5843 | 0.4050 | - |
| | Top-Bot | 0.1809 | 0.5930 | **0.6530** | 1.8% | 0.1800 | 0.5996 | 0.4300 | 3.1% |
| | Mid-1/3 | **0.1877** | **0.5963** | 0.6510 | **2.3%** | **0.1860** | **0.6034** | **0.4380** | **4.7%** |
| | Mid-2* | **0.1877** | **0.5963** | 0.6510 | **2.3%** | **0.1860** | **0.6034** | **0.4380** | **4.7%** |
| **410M** | Pythia | 0.1852 | 0.6376 | 0.7400 | - | 0.1988 | 0.6415 | 0.4850 | - |
| | Top-Bot | 0.2031 | 0.6398 | 0.7370 | 1.1% | 0.2073 | 0.6464 | 0.4800 | 0.6% |
| | Mid-1/3 | **0.2108** | **0.6458** | 0.7400 | **2.2%** | **0.2125** | 0.6420 | 0.4760 | 0.4% |
| | Mid-2 | 0.2031 | 0.6398 | **0.7470** | 1.7% | 0.2039 | **0.6518** | **0.4870** | **1.3%** |
| **2.8B** | Pythia | 0.2253 | 0.6768 | 0.7910 | - | 0.2346 | 0.6844 | 0.4810 | - |
| | Top-Bot | 0.2270 | 0.6806 | 0.7920 | 0.4% | 0.2321 | 0.6866 | 0.5010 | 1.4% |
| | Mid-1/3 | 0.2253 | **0.6899** | 0.7890 | 0.7% | 0.2381 | **0.6997** | **0.5040** | **3.0%** |
| | Mid-2 | **0.2304** | 0.6795 | **0.8150** | **1.9%** | **0.2415** | 0.6817 | 0.4910 | 1.0% |

better performance compared to the baseline, the 2% inhibition level yields the best results, which is consistent with the pattern found in Subsection 4.2.

Additionally, we observe that a higher level of inhibition is preferred as model scale increases. Specifically, the 2% inhibition level is optimal for the 410M model, while 1% and 3% are best for the 70M and 2.8B models, respectively. This aligns with our analysis in Section 3, which suggests that larger models are more polysemantic and thus require more inhibition to improve performance.

### B.4 EFFECTIVE IN DECREASINIG MONOSEMANTICITY

To validate our L2E indeed inhibit the monose­manticity, we further analyze related dynamics of the Pythia models (Biderman et al., 2023) with and without L2E. Ideally, L2E will re­sults in lower monosemanticity, thus smaller MS scores. As we apply L2E in the mid­dle 2 layers, we inspect their threshold of top-2% MS, shown in Figure 12, where the upper layer is denoted as "Upper" and "Lower" for the lower layer.

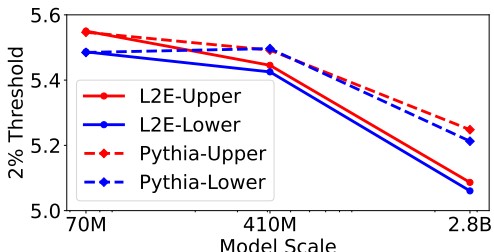

Figure 12: Threshold of top-2% MS when training with and without L2E (dash lines).

One can observe that the L2E method effec­tively reduces the monosemanticity of the mod­els, especially for larger models. Besides, we observe that the monosemanticity of the Pythia models decreases as the model scale increases. This aligns with our hypothesis that larger models are more polysemantic.

### B.5 INHIBITION ON DIFFERENT LAYERS

Our in-depth analysis of monosemanticity across layers in Section 3 and Section B.2 suggests that middle layers are less monosemantic. These layers are thought to handle complex reasoning by processing composed and abstract features. Consequently, our main experiments focus on inhibiting these middle layers.

To further investigate the impact of inhibiting different layers on model performance, we conducted additional empirical experiments. Three settings of L2E inhibition are tested on the Pythia models (Biderman et al., 2023): middle 2 layers (Mid-2), top and bottom layers (Top-Bot), and middle 1/3 layers (Mid-1/3). For instance, in the 24-layer 410M model, Mid-2 inhibits the 11th and 12th layers, Top-Bot inhibits the 1st and 24th layers, and Mid-1/3 inhibits the 8th to 16th layers.

Table 6: The results of L2E on ARC (Clark et al., 2018).

| Setting | | 0-shot | | | 5-shot | | |
|---|---|---|---|---|---|---|---|
| Scales | | 70M | 410M | 2.8B | 70M | 410M | 2.8B |
| **Arc-Easy** | Pythia | **0.3880** | **0.4545** | **0.5223** | 0.3708 | **0.4811** | **0.5522** |
| | L2E | 0.3813 | 0.4423 | 0.5042 | **0.3864** | 0.4798 | 0.5358 |
| **Arc-Challenge** | Pythia | 0.1706 | 0.1852 | 0.2304 | 0.1834 | 0.1988 | 0.2346 |
| | L2E | **0.1877** | **0.2031** | **0.2304** | **0.1860** | **0.2056** | **0.2415** |

Table 7: The performance when our Moving Threshold is applied or removed (No-Thr).

| Setting | | 0-shot | | | | 5-shot | | | |
|---|---|---|---|---|---|---|---|---|---|
| Datasets | | ARC-C | PIQA | SciQ | ↑ | ARC-C | PIQA | SciQ | ↑ |
| **70M** | Pythia | 0.1706 | 0.5887 | 0.6430 | - | 0.1834 | 0.5843 | 0.4050 | - |
| | No-Thr | 0.1792 | 0.5914 | 0.6430 | 0.8% | 0.1741 | 0.5925 | 0.4110 | 4.1% |
| | L2E | **0.1877** | **0.5963** | **0.6510** | **2.3%** | **0.1860** | **0.6034** | **0.4380** | **4.7%** |
| **410M** | Pythia | 0.1852 | 0.6376 | 0.7400 | - | 0.1988 | 0.6415 | 0.4850 | - |
| | No-Thr | **0.2048** | 0.6360 | **0.7470** | 1.6% | **0.2056** | 0.6431 | 0.4730 | -0.3% |
| | L2E | 0.2031 | **0.6398** | **0.7470** | **1.7%** | 0.2039 | **0.6518** | **0.4870** | **1.3%** |
| **2.8B** | Pythia | 0.2253 | 0.6768 | 0.7910 | - | 0.2346 | 0.6844 | 0.4810 | - |
| | No-Thr | 0.2159 | **0.6866** | 0.8060 | 0.9% | 0.2406 | **0.6921** | 0.4590 | -0.6% |
| | L2E | **0.2304** | 0.6795 | **0.8150** | **1.9%** | **0.2415** | 0.6817 | **0.4910** | **1.0%** |

Table 5 presents our results. Consistent with our analysis, inhibiting the middle layers yields the best performance improvement, with Mid-2 and Mid-1/3 achieving the highest scores in almost all cases (except for Top-Bot with 70M on 0-shot SciQ). Mid-2 would be the more efficient setting while maintaining similar performance. However, these finding opens up intriguing points for further research. For example, given that different layers may play different roles in a model's capabilities, inhibiting specific layers might benefit particular tasks.

## B.6 SOME SPECIAL RESULTS

Besides the datasets tested in the main experiments, we also obtained some noteworthy results on other datasets. ARC-Easy and ARC-Challenge are two partitions of data from (Clark et al., 2018), with ARC-Easy being the simpler set. While conducting experiments, we discovered negative results on ARC-Easy from L2E, as shown in Table 6. Interestingly, the results were remarkably consistent: L2E consistently outperformed the original model on ARC-Challenge but rarely improved on ARC-Easy. This pattern leads to a hypothesis that L2E may be more effective on more challenging tasks. This aligns with (Wang et al., 2024), which suggests that monosemanticity functions like hard memorization—potentially impairing performance on complex tasks while being beneficial for simpler ones. Some studies on grokking also treat memorization as a negative pattern when dealing with complex mathematical problems (Liu et al., 2022), while (Yan et al., 2024) also finds some positive impacts of monosemanticity. The potential influence of monosemanticity is still in its early stages of exploration.

## B.7 ABLATION STUDY ON MOVING THRESHOLD FOR RETRIEVAL

Recall that we introduced a moving threshold to enable efficient computation and capture a global impact. As shown in Table 2, this approach nearly eliminates the additional computational cost introduced by MEmeL.

Here, we further investigate the impact of the moving threshold on model performance. We conducted an ablation study, replacing the moving threshold with the original sorting method in MEmeL. The results are presented in Table 7. Our moving threshold consistently outperforms the fixed threshold, likely due to its ability to obtain global statistics of monosemanticity.

Table 8: The performance of our regularization-style inhibition compared with the Reverse Deactivation (RD) (Wang et al., 2024).

| Setting | | 0-shot | | | | 5-shot | | | |
|---|---|---|---|---|---|---|---|---|---|
| Datasets | | ARC-C | PIQA | SciQ | ↑ | ARC-C | PIQA | SciQ | ↑ |
| **70M** | Pythia | 0.1706 | 0.5887 | 0.6430 | - | 0.1834 | 0.5843 | 0.4050 | - |
| | RD | **0.1894** | 0.5424 | 0.4460 | -16% | **0.1928** | 0.5495 | 0.3170 | -9.7% |
| | L2E | 0.1877 | **0.5963** | **0.6510** | **2.3%** | 0.1860 | **0.6034** | **0.4380** | **4.7%** |
| **410M** | Pythia | 0.1852 | 0.6376 | 0.7400 | - | 0.1988 | 0.6415 | 0.4850 | - |
| | RD | 0.1962 | 0.6099 | 0.6430 | -7.3% | 0.1809 | 0.6235 | 0.4480 | -5.5% |
| | L2E | **0.2031** | **0.6398** | **0.7470** | **1.7%** | **0.2039** | **0.6518** | **0.4870** | **1.3%** |
| **2.8B** | Pythia | 0.2253 | 0.6768 | 0.7910 | - | 0.2346 | **0.6844** | 0.4810 | - |
| | RD | 0.2133 | 0.6513 | 0.7680 | -3.6% | 0.2244 | 0.6600 | 0.4670 | -3.5% |
| | L2E | **0.2304** | **0.6795** | **0.8150** | **1.9%** | **0.2415** | 0.6817 | **0.4910** | **1.0%** |

### B.8 ABLATION STUDY ON THE REGULARIZATION-STYLE INHIBITION

To address the potential ineffectiveness of Reverse Deactivation (RD) (Wang et al., 2024) in pre-training, where its assumptions may not strictly hold, we proposed a regularization-style method to inhibit selected monosemantic neurons.

In this subsection, we conducted an ablation study comparing our method with RD, with results shown in Table 8. Our method outperformed RD, which is consistent with our analysis.

However, it's important to note that our current experiments only train the model for 10% of the total steps, which is a setting that favors our approach. As the model becomes well-trained, RD's assumptions may become valid, potentially increasing its effectiveness. The optimal selection or combination of inhibition methods remains an area for further exploration.

## C DISCUSSION

In addition to the discussions on mechanistic interpretability, information bottleneck, existence of emergence, biological perspectives, and brain-inspired learning in Wang et al. (2024), we present further discussions on related important deep learning topics in this section.

### C.1 RELATIONSHIP WITH GROKKING

While monosemanticity functions like rote memorization in neural networks, a phenomenon called "grokking" also explores the negative impact of neuron memorization. The initial study by Power et al. (2022) observed grokking in very small models, where the test loss dramatically decreased after the training loss had converged for a long time. The authors highlighted an under-explored area: how neural networks generalize beyond mere memorization? Unlike studies on emergence, their research meticulously examined very small models (with only 2 layers). Interestingly, their analysis on algorithmic datasets aligns with our findings, hypothesizing that memorization hinders more complex reasoning.

To investigate the reason of grokking, Liu et al. (2022) examines multiple datasets using models with fixed L2 norm of weight $w_c$. They find that a large $w_c$ leads to poor representation and impairs generalization in grokking. On the other hand, Nanda et al. (2023) focuses on the modular addition task and reverse engineering the model weights. They summarize the training process leading to grokking in 3 stages: memorization, circuit formation, and memorization component removal. Notably, both papers highlight grokking's potential to aid in understanding emergence. However, unlike our work, their studies were limited to very small models.

Subsequent research has conducted case studies to explore the benefits of grokking (Xu et al., 2024) and gain deeper understanding (Rubin et al., 2024; Levi et al., 2024; Kumar et al., 2024). Notably, Levi et al. (2024) raises doubts about whether the occurrence of grokking might be due to the accuracy measure used, a concern similar to that raised about Emergence (Schaeffer et al., 2023).

To conclude, complementing our research, grokking-related studies provide highly detailed analyses of small models (Liu et al., 2022; Doshi et al., 2024; Pearce et al., 2023), similar to observing particle motion through a microscope. Hopefully, these two research directions could yield discoveries from different perspectives and potentially combine to further enhance model performance, such as interchanging techniques for inducing grokking and emergence (Wang et al., 2024; Lyu et al., 2024). Besides, we highlight an interesting example from their experiments, where an extremely small model (with only 5 neurons) is incapable of even memorizing (Pearce et al., 2023). This raises a question: for current challenging tasks, where do our models stand? Are they at the stage of partial memorization awaiting generalization, or have they not even reached the point of memorization?

## C.2 RELATIONSHIP WITH SPARSITY

When we introduce the idea of inhibiting monosemanticity, many readers express concern about its influence on sparsity, which is useful for efficient computation and model compression, especially in large models. In this section, we provide a concise review of sparsity and its potential conflicts and cooperation with our proposition.

As models grow larger, researchers have observed that many weights become near-zero or insignificant, forming sparse connections between neurons (Han et al., 2015b; Li et al., 2017; Han et al., 2015a). Leveraging this sparsity, various pruning strategies for weights and neurons have been developed to compress models while maintaining performance. Although weight-based pruning is theoretically efficient (Frankle & Carbin, 2019), it presents practical challenges (Sun et al., 2021), often resulting in irregular networks that are difficult to deploy. To address this, new GPU architectures have been proposed to support sparse computation (NVIDIA, 2020), alongside methodological improvements (Zhou et al., 2021; Liu et al., 2021; Frantar & Alistarh, 2023).

In addition to weight sparsity, researchers also focus on activation sparsity in neural networks (Kurtz et al., 2020; Akiva-Hochman et al., 2022). As discussed in Section 2, when using the ReLU activation function, neuron outputs $\leq 0$ are naturally considered inactive, creating a form of sparsity. Researchers leverage this sparsity to reduce computation and enhance it further by setting values below a higher threshold to 0 (Kurtz et al., 2020). Combining both weight and activation sparsity can improve model efficiency (Akiva-Hochman et al., 2022). Unlike static weight sparsity, utilizing activation sparsity requires monitoring dynamic flow (i.e., each input has a different sparsity pattern), which is more challenging and functionally similar to our L2E.

In recent years, as large language models have grown, high-level designs supporting activation sparsity have emerged, such as the Mixture-of-Experts (MoE) layer (Fedus et al., 2022). While pruning based on activation values induces sparsity at irregular positions, MoE activates only one expert for each input, creating physical continuity on the GPU. This approach is more hardware-friendly and thus preferred in industrial pipelines.

The main conflict between sparsity and our method lies in the functional assumption of activation. Monosemanticity forms a 1-to-1 correlation, which is considered similar to rote memorization and may hinder complex reasoning abilities. However, this 1-to-1 mapping is favored in sparsity studies. When we reduce monosemanticity, a single input would activate multiple neurons, rendering pruning based on sparsity ineffective. If both approaches are proved valid, balancing L2E and sparsity will become a trade-off between effectiveness and efficiency.

Fortunately, our current studies focus on the inhibition of **extremely** monosemantic neurons, which can potentially co-exist with sparsity. We demonstrate the ideal case in Figure 13, where highly monosemantic neurons are inhibited (brown), while a large number of inactive neurons can be pruned (green) using sparsity methods. Besides, when using MoE, pruning occurs during expert selection, while each expert can be further enhanced with our method (i.e., made more polysemantic). Additionally, the MS metric could serve as a supervisor for pruning unimportant neurons to achieve sparsity. These potential collaborations await future research.

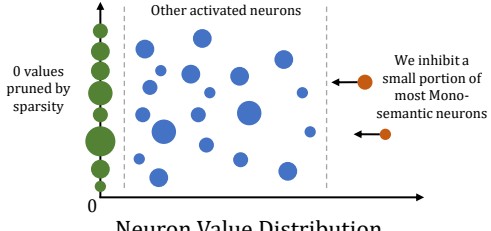

Figure 13: Ideal distribution of neuron values. Pruning based on sparsity and inhibiting monosemanticity can coexist.

