# OpenReview forum: "Learning Towards Emergence: Paving the Way to Induce Emergence by Inhibiting Monosemantic Neurons on Pre-trained Models"
_ICLR.cc/2025/Conference — Submitted to ICLR 2025_

### Official Review · Reviewer_nf4t · 2024-11-03

**Soundness:** 2
**Presentation:** 2
**Contribution:** 2
**Rating:** 5
**Confidence:** 3

**Summary:**

The manuscript presents an approach termed Learning Towards Emergence (L2E), which aims to enhance the performance of large pre-trained neural networks by inhibiting monosemantic neurons. The authors argue that as model scale increases, the presence of monosemantic neurons diminishes, potentially contributing to the phenomenon of emergence. L2E employs the FKR, an efficient retrieval method, and proposes a regularization-style inhibition approach. The experiments demonstrate the effectiveness of L2E in improving model performance during pre-training.

**Strengths:**

* The introduction and the related works are clear.
* The motivation is straightforward that previous methods for inhibiting monosemantic neurons lack efficiency and experimental evidence.

**Weaknesses:**

* The study is limited to Pythia models, and it is unclear how well L2E would perform on other architectures or datasets.
* Table 1 lacks the comparison results of related works, such as MEmeL.
* While the introduction highlights the potential of suppressing monosemantic neurons to promote emergence, the experiments presented in this paper fall short of validating this hypothesis.

**Questions:**

1. The y-axes in Table 1(a) and 1(b) are inconsistent. It is unclear why the minimum value in Table 1(a) is 2 rather than -2.
2. The authors claim that the neurons in Table 1(b) are polysemantic, yet the data suggests that they are inactive to all inputs.

---

### Official Review · Reviewer_whwH · 2024-11-06

**Soundness:** 2
**Presentation:** 3
**Contribution:** 2
**Rating:** 3
**Confidence:** 4

**Summary:**

This paper investigates the relationship between model scaling and monosemantic neurons, introducing a novel method called Learning Towards Emergence (L2E). The key observation driving this research is that larger language models tend to exhibit fewer monosemantic neurons - neurons that form one-to-one mappings with interpretable features. Building on this insight, the authors develop L2E to actively inhibit monosemantic neurons during the pre-training phase of large models. The method's effectiveness was validated on Pythia models ranging from 70M to 2.8B parameters. Through comprehensive experiments, they demonstrate that inhibiting monosemantic neurons can enhance model performance across multiple downstream tasks. A particularly significant contribution is their efficient implementation using adjustable thresholds, which addresses the computational challenges in scaling to larger models. The authors also introduce the False Killing Rate metric to quantify inhibition effects. While the study is currently limited to the Pythia model family, it provides valuable insights into neural network organization at scale and suggests a potential mechanism for understanding emergence phenomena in large language models. The findings open new avenues for improving pre-training strategies, though further research is needed to fully validate the connection to emergence.

**Strengths:**

1.The study of emergence in large language models is a compelling and significant research direction. Understanding why and how models suddenly exhibit enhanced capabilities beyond simple scaling laws provides crucial insights into the nature of large language models. This kind of research is fundamental for both advancing our theoretical understanding and guiding practical model development.

2.This paper provides fresh perspectives and introduces novel tools for understanding emergence, including the False Killing Rate metric and adjustable thresholding techniques. By linking monosemantic neurons to model performance, it offers a new angle for analyzing how neural networks organize and process information as they scale up, while also providing practical methods to enhance model capabilities.

3.The experimental results are comprehensive and convincing, demonstrating consistent performance improvements across various model scales (70M to 2.8B) and downstream tasks. The proposed L2E method not only shows effectiveness in improving model performance but also addresses efficiency challenges in large-scale implementation, making it practically viable for real-world applications.

**Weaknesses:**

1.The paper's experimental scope falls significantly short of the scale where emergence typically occurs. Current observations suggest that dramatic emergence phenomena primarily manifest in models around 70B parameters or larger, where there's a clear performance gap compared to smaller models. By limiting their analysis to models up to 6.9B parameters, the study misses the most critical scale range where emergence actually happens. This substantially weakens their conclusions about emergence and raises questions about the practical applicability of their findings to truly large-scale models.

2.The research is exclusively conducted on the Pythia model family, and the results are heavily dependent on the quality of feature datasets. For a study claiming to understand fundamental properties of large language models, this narrow focus is concerning. A more comprehensive analysis should include widely-used open-source models like Llama, BLOOM, or Falcon, which have different architectures and training approaches. This would help validate whether their findings about monosemantic neurons are truly general properties or just specific to Pythia models.

3.Without analyzing mainstream models, the paper's conclusions remain largely theoretical and potentially disconnected from practical reality. While acknowledging the challenges of training large-scale models, the authors could have conducted analysis and interpretation on existing pre-trained models. This would have provided valuable validation of their hypotheses and strengthened their arguments about the relationship between monosemantic neurons and emergence. The lack of such analysis leaves a significant gap between their theoretical framework and real-world applications.

**Questions:**

1. How do you define and measure emergence more precisely? The paper suggests a connection between monosemantic neurons and emergence, but is this correlation or causation?

2. What is the theoretical justification for why reducing monosemantic neurons would lead to better model performance?

3. Why choose 2% as the optimal threshold for neuron inhibition? Could this vary with model size or architecture?

4. How robust is the False Killing Rate metric across different types of models and tasks?

5. Could the efficiency improvements in L2E potentially sacrifice some accuracy in identifying monosemantic neurons?

6. Given that emergence typically occurs in models around 70B parameters, how relevant are the findings from much smaller models (up to 6.9B)?

7. Why focus exclusively on the Pythia model family? How would the results generalize to other architectures?

8. What specific characteristics of the feature datasets might affect the reliability of the results?

9. Could this method be applied to existing pre-trained models without retraining?

10. How would this approach scale to truly large models where emergence is actually observed?

11. What are the computational costs and practical challenges of implementing L2E in production environments?

12. How might this work extend to understanding other emergent properties in large language models?

13. Could this approach be adapted for other types of neural networks beyond language models?

14. What additional metrics or methods could help validate the connection between monosemantic neurons and model performance?

---

> ### Comment · Reviewer_whwH · 2024-12-01
>
> Dear Authors:
>
> I have reviewed the updated PDF and the overall comments provided by the authors. Since there was no specific rebuttal addressing the concerns I raised in my initial review, I maintain my original rating.
>
> Best regards

---

> > ### Author Response · Authors · 2024-12-01
> >
> > Dear reviewer whwH,
> >
> > Thank you for your comments and attention.
> >
> > The major concerns about emergence and model selection are currently unresolvable, which had been discussed in Section 5.4 (Limitation) of our first submission and also in our ICLR reply. We clearly acknowledge this gap, which is not supposed to be (and cannot be) resolved in this paper.
> >
> > We greatly cherish your and all the reviewer's comments, and we are adding experiments for effectiveness and hypothesis validation. We will also clarify our contribution more precisely to avoid overstatement, and possibly make it a journal submission instead as strong extension.
> >
> > Thank you again for your valuable feedback and openness to further discussion. We hope to make it publicly available first, so we have not withdrawn the paper, worrying whether this will influence its availability. Hope this can make people working in this direction benefit from both our paper and your insightful comments.
> >
> > Best regards

---

### Official Review · Reviewer_fXs7 · 2024-11-10

**Soundness:** 2
**Presentation:** 3
**Contribution:** 2
**Rating:** 3
**Confidence:** 4

**Summary:**

This authors conduct extensive analysis on monosemanticity. They further upgrade previous methods with more carefully determined number of inhibited neurons based on False Killing Rate, a moving threshold for global inhibition with better efficiency, and a regularization term for models in the early pre-training stage. The proposed method surpasses MEmeL and the backbone model in both efficiency and effectiveness.

**Strengths:**

1. The paper is well organized and is easy to follow.
2. The paper includes a thorough overview of the background and development of prior arts.
3. The authors provide interesting analysis on monosemantic neurons, and extend the experiments to relatively larger models. The insights can be useful to works in this direction.

**Weaknesses:**

1. The experiments are not strong enough to support the claims.
(a) Since the title suggests this is a study towards "emergence", experiments should at least cover commonly used benchmarks (e.g. BIG-Bench, MMLU, etc.), at least the ones used in the original MEmel paper should be included for fair comparison. The motivation of benchmarks should be carefully stated in the paper.
(b) The proposed methods do not use any assumption from natural language settings, hence they should work for tasks from other domains as well (e.g. vision). Benchmarks from other domains are not necessary but can be very helpful to support the effectiveness of the proposed methods.
(c) The improvement in performance is somewhat marginal. Given the scale of improvement, it will be more convincing if signifcance tests can be provided.
(d) The ablation study in the appendix should be moved to Section 5 with more serious discussion.

2. I enjoy reading the analysis part, but unfortunately the technical novelty is rather limited in this work. The three major tricks may be useful, but are somehow heuristic and lack support from more rigorous experiments and ablation study (as stated in the first point). Without such analysis we can hardly determine which part of the ingredient works, and whether it is an ad-hoc improvement by accident or it can be generalized to other scenarios.

3. I find the title "learning towards emergence" can be misleading. The previous work "learning from emergence" has clear motivation from the relationship between emergence and monosemantic neurons. However, one would expect "learning towards emergence" can promote models with relatively small scale to exhibit emergence similar to large models. It is a bit disappointing to see the work is some improvement on existing methods within the same framework.

**Questions:**

1. Can this method be generalized to other domains and other benchmarks?
2. From Figure 6, the optimal percentage seems to be 1%. From Table 4, neither 1% nor 2% achieves consistently better performance (not to mention significance). Is there a more robust way for this percentage selection?
3. Will the authors add more analysis on the relationship between the proposed method and "emergence"?

---

### Author Response · Authors · 2024-11-13

We sincerely thank all the reviewers for their accurate and valuable comments. We highlight two common questions raised about this work:

1. No experimental emergence, which commonly occurs in models with sizes ≥70B.
2. Limited to Pythia models only.

After working on this topic, we engaged with companies and learned how large models exhibiting emergence are developed: extensive manual, pipeline-style parallel trial and error (incrementally training on multiple small data subsets based on the current checkpoint, selecting the best for the next round). Even if we could support computational resources equivalent to 70B parameter models, without a very large team, it's almost impossible to train a model capable of emergence in a single attempt.

So we chose Pythia, which provides complete datasets and training data order, thus reducing the gap in resources and manpower during training. This also allows us to demonstrate, based on available computing power: 1) improvements in a widely used model under pre-training settings, and 2) a (possible) full-scale training comparison if more funding becomes available.

This work thoroughly demonstrates everything we can accomplish with our current resources: effective and necessary efficiency improvements for future (potential) complete pre-training; and validation experiments based on classic monosemantic analysis tools like probing. Many experiments and analyses serve as purely instructional guides—helping users avoid pitfalls and even assisting potential competitors in accomplishing our ideas on emergence (if they have sufficient resources).

We acknowledge and clearly understand that researching emergence in ordinary research institutions is a fragile dead-end (and that’s why we name the paper “towards emergence”); we're simply contributing as much as we can before we burn out.

&nbsp;

We've further added three pages of appendices after submission and now available above, particularly discussing two related directions (grokking and sparsity). More experiments in B.8 address the baseline concerns raised by Reviewers nf4t and fXs7. Hope this will be useful for those who are concerned.

---

> ### Comment · Reviewer_fXs7 · 2024-12-01
>
> Thank you for the response. I will keep my original rating.

---

> > ### Author Response · Authors · 2024-12-01
> >
> > Dear reviewer fXs7,
> >
> > Thank you for your comments and glad to see your reply.
> >
> > Your comments are very helpful and actionable. We worked on adding MMLU and Big-bench evaluations recently. Since the Big-bench dataset is quite large (over 200 tasks), we are using their recommended BIG-bench Lite (BBL) dataset, which contains 24 tasks.
> >
> > We have completed testing on MMLU and also evaluations of BBL on the 70M and 410M models. Below is a snapshot of our results.
> >
> > Results of MMLU:
> > |  Setting | |0-shot |||5-shot |  |
> > | :---: | :---: | :---: | :---: | :---: | :---: | :---: |
> > | Scales | 70M| 410M | 2.8B | 70M| 410M | 2.8B |
> > | Pythia| **0.2311**| 0.2303| 0.2306 |0.2438| 0.2470 |0.2432|
> > | **L2E** | 0.2301| **0.2306** |**0.2534**| **0.2497** |**0.2472** |**0.2443**|
> >
> > Results of BBL:
> > |  Setting | |0-shot |||5-shot |  |
> > | :---: | :---: | :---: | :---: | :---: | :---: | :---: |
> > | Scales | 70M| 410M | 2.8B | 70M| 410M | 2.8B |
> > | Pythia| 0.2188 |0.2198| run |0.2140 |**0.2377**| run|
> > | **L2E** | **0.2242**| **0.2199**| **run**| **0.2235**| 0.2266| **run**|
> >
> > More discussions of the results are also added. We also gave the descriptions of used datasets in the current paper. If you are interested in it, I can prepare an anonymous link for you.
> >
> > Best regards

---

### Meta-Review · Area_Chair_yKwA · 2024-12-16

**Metareview:**

This paper presents an algorithm to facilitate the emergent abilities in large language models. It works by inhibiting the monosemantic neurons and thus improving the zero-shot and few-shot accuracy of LLMs. However, as the reviewers point out, the studied LLMs are quite small, which is far insufficient to validate the algorithm's effectiveness and convince the reviewers (and AC) that the algorithm can scale up to proper-sized LLMs. The reviewers raised a lot of questions, but the authors did not address them carefully in the rebuttal, and the reviewers confirmed that they would like to keep the original (negative) ratings. Finally, with a unanimous recommendation (3/3/5), the AC follows the reviewers to reject this submission.

**Additional Comments On Reviewer Discussion:**

The reviewers raised a lot of questions, but the authors did not answer them one-by-one, even providing unlimited space for rebuttal. The reviewers chose to keep the original scores -- rejection.

---

### Decision · Program_Chairs · 2025-01-22

Reject